# Propuesta de Trabajo de Investigación: Optimización Simheurística para la Flexibilidad Energética Considerando Variables Estocásticas en Sistemas Renovables

## Abstract

Este proyecto propone el estudio de un modelo de optimización simheurística para la planificación de la gestión energética en sistemas con generación renovable y almacenamiento, considerando la incertidumbre inherente en los modelos predictivos de generación, demanda y precios del mercado. El algoritmo combinará búsqueda heurística *Branch & Bound* con metaheurísticas de búsqueda local y aleatorización sesgada, integrando la simulación de diferentes escenarios mediante técnicas Monte Carlo. El objetivo es minimizar costes operativos y reducir riesgos en un horizonte de 48 horas, equilibrando calidad de solución y viabilidad computacional. El algoritmo se validará con un conjunto de datos sintético basado en parámetros de diferentes sectores industriales y se comparará frente a enfoques deterministas, empleando técnicas bayesianas para un análisis estadístico robusto. Se espera que los resultados proporcionen estrategias más eficientes en la gestión energética en sectores con alta penetración renovable, reduciendo los costes de operación y el impacto ambiental. Este proyecto se desarrollará en el marco de un programa de doctorado, relacionado con la optimización de la flexibilidad energética.

## 1   Introducción

En los últimos años se han logrado avances en la generación de energía eléctrica renovable, en particular eólica y solar, y en el almacenamiento energético mediante baterías, aumentando la eficiencia de estos sistemas. La combinación de ambas tecnologías, unida a la fluctuación de los precios en el mercado diario, ofrece oportunidades para incrementar la flexibilidad energética que resultan especialmente relevantes en industrias con altos consumos de electricidad. Una planificación optimizada de la gestión de la producción y el almacenamiento puede servir para determinar los momentos óptimos de compra y vertido a red, mejorando el rendimiento económico, la estabilidad de los sistemas y además reducir el impacto ambiental.

Sin embargo, la incertidumbre inherente en la predicción de la generación renovable, en la demanda y la variabilidad en los precios pueden afectar significativamente a dicha planificación y a los rendimientos obtenidos, sobre todo a largo plazo. Los enfoques tradicionales de programación determinista no permiten modelar adecuadamente el carácter estocástico de dichas estimaciones, lo que puede derivar en soluciones subóptimas o económicamente inviables si finalmente se dan condiciones adversas, no tenidas en cuenta por los modelos predictivos. Por lo tanto, resulta necesario aplicar otras metodologías que incorporen dicha variabilidad, de modo que las planificaciones

Submitted to XVI Congreso Español de Metaheuristicas, Algoritmos Evolutivos y Bioinsiprados (MAEB 2025). Do not distribute.

recomendadas proporcionen tanto una minimización de los costes como una reducción de los riesgos en situaciones desfavorables.

En este proyecto de investigación se propone un enfoque de optimización simheurística, combinando métodos de simulación para evaluar diferentes escenarios posibles con algoritmos de optimización, que establecerán una planificación robusta ante diferentes situaciones en un horizonte temporal de 48 horas. El objetivo es mejorar la toma de decisiones al considerar la aleatoriedad de las variables involucradas debida a los errores de estimación de los modelos predictivos. El algoritmo propuesto combinará metodologías de ramificación y poda (*Branch & Bound*) con optimización metaheurística para la búsqueda local. Además, se aplicarán técnicas de aleatorización sesgada (*biased randomization*), para mejorar la exploración del espacio heurístico. La simulación de escenarios mediante técnicas Monte Carlo (MC) permitirá estimar los costes esperados bajo diferentes desviaciones en los pronósticos. Se espera que la combinación de estas técnicas proporcione una exploración más diversa de soluciones factibles, permitiendo minimizar los costes de operación y, a su vez, mejorar la robustez de la estrategia de gestión energética.

## 2  Motivación

Los enfoques simheurísticos y de optimización estocástica han sido ampliamente estudiados en sectores como la logística y la salud [1, 2], pero su aplicación en flexibilidad energética sigue siendo limitada en la literatura actual. Dado el crecimiento de las energías renovables y el impacto de la incertidumbre en su gestión, la integración de estos métodos en la planificación energética representa una oportunidad de investigación con un alto potencial de impacto, tanto desde una perspectiva industrial como ambiental.

Desde un punto de vista práctico, la planificación energética debe equilibrar la optimización de costes con la capacidad de reacción ante condiciones adversas. Por otro lado, el tiempo de cómputo no debe ser excesivo para ser aplicable en entornos de operación reales. Por tanto, la solución proporcionada debe ofrecer un compromiso entre el tiempo de cómputo, la calidad de las soluciones y la reducción del riesgo ante situaciones desfavorables [3].

Para mejorar la eficiencia computacional, este estudio explorará estrategias avanzadas de poda y heurísticas que permitan una exploración inteligente del espacio de búsqueda [4]. En particular, se evaluarán tres estrategias:

- Aleatorización sesgada (*biased randomization*) para diversificar la exploración y evitar estancamientos en óptimos locales [5].
- Eliminación de ramas con baja probabilidad de contener soluciones óptimas basándose en análisis estadístico de Monte Carlo [6].
- Poda de ramas con alto riesgo de escenarios desfavorables, priorizando soluciones con mayor estabilidad frente a otros más imprevisibles [7].

Para garantizar la aplicabilidad en entornos reales, se desarrollará un conjunto de datos sintético basado en parámetros industriales, tomando como referencia informes institucionales y auditorías energéticas realizadas por expertos. El rendimiento del algoritmo propuesto se comparará frente a enfoques deterministas y exclusivamente heurísticos o metaheurísticos.

## 3  Hipótesis

Este estudio se basa en la premisa de que la incorporación de la incertidumbre en la optimización de la programación energética mejorará la eficiencia en costes y reducirá la variabilidad de los resultados, especialmente cuando se considere su aplicación en el largo plazo. Las hipótesis principales son las siguientes:

1. La incorporación explícita de la incertidumbre en las decisiones de planificación mejora la eficiencia en costes a largo plazo, ya que permite una mejor evaluación de los riesgos [8].
2. Un enfoque híbrido que combine *Branch & Bound* con metaheurísticas de búsqueda local mejora la calidad de las soluciones frente a modelos exclusivamente heurísticos o metaheurísticos, al aprovechar poda estructurada y una exploración más eficiente. Se espera que

esta aproximación proporcione mejor equilibrio entre el tiempo de cómputo y las soluciones alcanzadas [3, 9].

3. La inclusión de técnicas de aleatorización sesgada y evaluación de escenarios mediante Monte Carlo garantizará la robustez de la planificación bajo condiciones finales inciertas, sin comprometer la viabilidad computacional [1, 4].

# 4 Objetivos

El objetivo de esta investigación es obtener una planificación óptima de la operación de sistemas fotovoltaico-eólicos con almacenamiento en un horizonte de 48 horas, minimizando los costes de operación y a su vez siendo robustos ante situaciones adversas. Para ello se desarrollará y evaluará un enfoque de optimización simheurística que integre *Branch & Bound* con metaheurísticas de búsqueda local para la planificación, incorporando la aleatoriedad en la predicción de la generación, los precios de mercado y la demanda. La granularidad de estas variables será horaria, dado que se considera suficiente para reflejar los cambios en las diferentes variables y es ampliamente utilizado en aplicaciones similares. Para alcanzar este objetivo, el estudio busca:

- Formular y diseñar un modelo de optimización simheurístico que integre poda estructurada y búsqueda local para mejorar la eficiencia en la programación.

- Explorar y evaluar estrategias de poda basadas en análisis estadístico de posibles soluciones, considerando tanto la poca probabilidad de escenarios óptimos como el riesgo de situaciones extremadamente desfavorables.

- Simular la variabilidad de las condiciones mediante técnicas Monte Carlo, de modo que se consiga realizar una evaluación adecuada de las distintas situaciones sin que ello suponga un aumento significativo en los tiempos de cómputo.

- Desarrollar un conjunto de datos sintético que represente errores de pronóstico en generación, demanda y precios de mercado.

- Aplicar técnicas de evaluación bayesiana para comparar de forma robusta el rendimiento del modelo propuesto con enfoques deterministas y modelos estocásticos donde no se combine búsqueda heurística con metaheurísticos.

# 5 Metodología

El primer paso de la investigación es la formulación del problema, definiendo el modelo matemático para la programación de un sistema fotovoltaico-eólico con almacenamiento, incluyendo restricciones de balance energético, operación del almacenamiento y participación en el mercado. También el diseño de la función objetivo, que debe minimizar los costes operativos totales, integrando costes de compra de energía, degradación del almacenamiento o penalizaciones por incumplimiento de la demanda.

A continuación, para garantizar la aplicabilidad del estudio, se generará un conjunto de datos sintético que refleje relaciones de parámetros reales en entornos industriales en España. Este conjunto de datos se diseñará con base en estudios previos actualmente en fase de aceptación.

Posteriormente se desarrollará el modelo de optimización simheurístico, integrando búsqueda discreta basada en *Branch & Bound* y aleatorización sesgada con métodos metaheurísticos para refinar la búsqueda local en el dominio continuo. La selección de los modelos metaheurísticos se realizará en base a la literatura, escogiendo aquellos reportados como especialmente eficientes en la fase de explotación en años recientes. Por otro lado, la simulación de escenarios posibles se realizará mediante técnicas *MC*.

A pesar de los beneficios esperados, el algoritmo enfrenta ciertos desafíos. La precisión de las simulaciones dependerá de la correcta calibración de los escenarios, lo que podría afectar la validez de los resultados si los datos de entrada no reflejan con fidelidad la incertidumbre real. Además, la combinación de simulación y optimización puede generar costos computacionales elevados en problemas de gran escala, lo que requerirá un equilibrio entre la exploración del espacio de búsqueda y la viabilidad computacional.

Una vez desarrollado el algoritmo, se procederá a su evaluación comparativa con modelos alternativos, incluyendo enfoques deterministas, heurísticos y metaheurísticos puros. Se analizará el desempeño del modelo propuesto en términos de calidad de solución, tiempo de cómputo y reducción de riesgos. En particular, se comparará el impacto de las estrategias de poda y simulación en la eficiencia computacional y la robustez de la planificación energética.

Finalmente, se realizarán experimentos computacionales para evaluar el desempeño de los modelos bajo distintos escenarios, cuantificando su eficiencia mediante técnicas bayesianas. Esta fase garantizará que las estrategias propuestas sean aplicables en entornos industriales reales y proporcionen soluciones óptimas en términos de coste y resiliencia operativa.

## 6 Conclusión

Esta investigación propone un modelo de optimización simheurística para la programación energética flexible bajo incertidumbre, combinando *Branch & Bound*, metaheurísticas y simulación Monte Carlo. Su rendimiento se comparará con el de otros enfoques deterministas y heurísticos mediante técnicas bayesianas, teniendo en cuenta la calidad de las soluciones obtenidas, la reducción de los riesgos y el coste computacional. Los resultados contribuirán al desarrollo de estrategias de programación más robustas en sistemas energéticos con alta penetración de renovables con almacenamiento.

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
