# OpenReview forum: "Optimización Simheurística para la Flexibilidad Energética Considerando Variables Estocásticas en Sistemas Renovables"
_MAEB/2025/Projects_Track — MAEB 2025 Proyectos_

### Official Review · Reviewer_gkFh · 2025-03-18
**Trabajo interesante aunque necesita algo más de justificación.**

**Rating:** 4
**Confidence:** 4

**Review:**

Me parece un trabajo interesante y que tiene bastante potencial en el campo que se aplica, gestión energética, pero que necesita algo más de justificación en algunos aspectos:
- El porqué de usar Branch and Bound en optimización en lugar de otras muchas técnicas existentes, es algo que debe ser explicado detalladamente.
- Algunos resultados deberían haberse mostrado y comentado brevemente porque, a pesar de la limitación de espacio, alguna tabla con sus comentarios posteriores vendrían bien.
En mi opinión, el trabajo debe ser aceptado aunque algunos aspectos pueden mejorar la calidad del trabajo o quizás la percepción.

---

### Official Review · Reviewer_wneX · 2025-03-19
**Un proyecto interesante, mejorar la motivación y usar datos reales**

**Rating:** 5
**Confidence:** 5

**Review:**

El proyecto plantea la optimización simheurística de sitemas de energía renovables.
En líneas generales el resumen está bien redactado y bien argumentado. Encuentro cuatro puntos en los que se podría mejorar el proyecto.
En  primer lugar el título me parece enrevesado, no veo la necesidad del extender tanto el mismo. Algo así como  "Optimización de sistemas de energía  renovables mediante metaheurísticas".

El segundo problema es el uso de la palabra simheurística como terminoogía. Conozco el término, pero no apruebo su uso. Una heurística es según la RAE una técnica de la indagación y del descubrimiento, o En algunas ciencias, manera de buscar la solución de un problema mediante métodos no rigurosos, como por tanteo, reglas empíricas, etc. Es decir, lo que proponen los autores es resolver el problema mediante una heurística (o metaheurística, que mejor para el MAEB). Para mí el uso de términos que quieren abrir campos nuevos dónde no los hay, es un error y recomiendo a los autores que reflexionen sobre este tema a la hora de presentarlo en MAEB.

En tercer lugar, creo que la motivación debería mejorarse, al menos con alguna referencia al estado del arte y sus carencias. No se justifica el uso de las metaheurísticas, simplemente se da por hecho que son necearias, pero no sabemos cuál es el motivo de tener que proponer una nueva técnica, seguro que lo hay, pero deberían explicarlo.

Por último está el uso de datos sintéticos. Si bien es normal el uso de datos sintéticos para comenzar el proyecto, no creo que sea correcto afirmar que de ahí se pueda aplicar a problemas reales de manera directa, ¿como se asegura esa traslación? los benchmarks deben estar muy bien diseñados para que esto sea sí. De nuevo una conclusión seria sobre este tema debería hacerse en la presentación en MAEB.

---

### Decision · Program_Chairs · 2025-03-20

Accept